# Feeling the Void: Lack of Support for Isolation and Sleep Difficulties in Pregnant Women during the COVID-19 Pandemic Revealed by Twitter Data Analysis

**DOI:** 10.3390/ijerph18020393

**Published:** 2021-01-06

**Authors:** Joey Talbot, Valérie Charron, Anne TM Konkle

**Affiliations:** 1Interdisciplinary School of Health Sciences, University of Ottawa, Ottawa, ON K1N 6N5, Canada; jtalb057@uottawa.ca; 2School of Psychology, University of Ottawa, Ottawa, ON K1N 6N5, Canada; vchar074@uottawa.ca; 3Brain and Mind Research Institute, University of Ottawa, Ottawa, ON K1N 6N5, Canada

**Keywords:** pregnancy, mental health, SARS-CoV-2, women’s health, social media

## Abstract

Pregnant women face many physical and psychological changes during their pregnancy. It is known that stress, caused by many factors and life events such as the COVID-19 pandemic, can negatively impact the health of mothers and offspring. It is the first time social media, such as Twitter, are available and commonly used during a global pandemic; this allows access to a rich set of data. The objective of this study was to characterize the content of an international sample of tweets related to pregnancy and mental health during the first wave of COVID-19, from March to June 2020. Tweets were collected using GetOldTweets3. Sentiment analysis was performed using the VADER sentiment analysis tool, and a thematic analysis was performed. In total, 192 tweets were analyzed: 51 were from individuals, 37 from companies, 56 from non-profit organizations, and 48 from health professionals/researchers. Findings showed discrepancies between individual and non-individual tweets. Women expressed anxiety, depressive symptoms, sleeping problems, and distress related to isolation. Alarmingly, there was a discrepancy between distress expressed by women with isolation and sleep difficulties compared to support offered by non-individuals. Concrete efforts should be made to acknowledge these issues on Twitter while maintaining the current support offered.

## 1. Introduction

Pregnancy is a period that comes with major life changes and many challenges. During the antenatal period of motherhood, approximately 84% of mothers will experience stress which can stem from preexisting medical conditions but also from sociocultural experiences; stress typically lowers as the pregnancy progresses [1]. Prenatal maternal stress can have varied impacts on the mother and the offspring, such as obstetric complications [2], low birth weight [3,4], delayed child development [5,6], postpartum depression, as well as other mental health problems [7]. Maternal stress is highly correlated to depressive symptoms, anxiety, and sleep difficulties [8,9]. Major negative life events during pregnancy can contribute to the mother’s stress, such as financial problems, death of a loved one, or global events like the COVID-19 pandemic [10].

This COVID-19 health crisis is caused by the severe acute respiratory syndrome coronavirus 2 (SARS-CoV-2), which was first reported in December 2019 [11]. Infection with this coronavirus is mainly characterized by respiratory difficulties as well as loss of smell and taste [11]. However, fever and a cough [12] have also been reported, as well as more chronic effects such as cognitive impairment, chronic fatigue, and chronic pain [13]. As of the end of June 2020, 10,357,662 people have reportedly been infected globally; this accounted to more than 508,055 deaths [14]. At the time of submission in December 2020, the numbers had increased to more than 61,000,000 cases of reported infection and over 1,400,000 deaths [15]. This virus is particularly dangerous for vulnerable populations, including pregnant women. Pregnancy complications can arise from infections that show similarities to COVID-19, such as influenza and severe acute respiratory syndrome (SARS) [16]. Previously published case studies report that pregnant women who develop viral pneumonitis stemming from a coronavirus infection have a higher mortality rate than non-pregnant women [16,17]. However, as of now, few studies have looked at the physiological and psychological consequences of the COVID-19 disease on pregnant women.

The lack of information with added uncertainty about the risks associated with this viral disease during pregnancy can trigger heightened stress about possible health effects for themselves or their offspring. It has been observed that during these unprecedented times, pregnant women tend to display more anxiety related to their health than prior to the COVID-19 pandemic [18]. Among the 71 pregnant women interviewed by Corbett and colleagues [18], over 63.4% expressed anxiety regarding their unborn child’s health and 66.7% expressed anxiety about their existing children. To prevent the spread of this virus and protect vulnerable populations, health authorities recommend common sanitary measures, including frequent hand washing, disinfecting surfaces with >60% ethanol, wearing face masks, and practicing social distancing [16,19]. However, being socially isolated for an extended period can increase feelings of boredom, loneliness, and stress, and cause distress, especially during a difficult situation like a global pandemic [20]. While some countries have slowly lifted restrictions regarding social distancing, vulnerable populations are still advised to stay isolated [16]. These precautionary isolation measures may have a significant negative impact on pregnant women, especially since their need for social support is great during this life changing period [21]. Pregnant women are more likely to experience symptoms of depression and anxiety if they are socially isolated or if they report low social support; a high level of social support during pregnancy has been linked to a reduced risk of antepartum depression [21,22]. Meanwhile, low levels of social contact are associated with a lower mood in women, especially linked to lower contact with friends and family [23]. Interventions to strengthen women’s social support during pregnancy may improve maternal well-being; one such intervention is online social support networks. A meta-analysis by [24] found that social media and mobile health applications can be effective to improve maternal well-being and reduce symptoms of depression and anxiety. Another study found that expecting women sought peer support in online health communities when their offline social network was not reliable, when they did not have access to healthcare professionals, or when they felt frustration towards their healthcare providers [25].

A netnographic approach to the use of social media can unravel much information on the state of self-reported mental health of pregnant women. Indeed, Twitter (www.twitter.com), a large internet-based social media website with over 186,000,000 daily active users, is popular for its ease of use, where users can share tweets in less than 280 characters to communicate statements [26]. Many studies have reported Twitter as a platform able to contribute to social support and information sharing on mental health [27,28]. Thus, Twitter can be used as a tool to assess the general state of events, such as the COVID-19 pandemic and its effect on the mental health of pregnant women and mothers [29]. There are multiple approaches to gather and analyze data from Twitter according to the research subject and keywords [30,31]. Some approaches utilize supervised machine learning to limit human error in finding tweets of interest [30]. Other approaches use automated data analysis and clustering techniques that allows for easy and efficient subject grouping [31]. This study used a keyword-driven data gathering tool and a lexicon-based analysis approach.

As of now, only a few studies have surfaced that examine the consequences of living through the novel coronavirus pandemic on the mental health of pregnant women. This is the first time we have access to such rich and diversified information coming from social networking platforms during a global pandemic; exploring its use by an international sample of individuals, companies, as well as medical professionals and researchers, will help us glean some insight into the support needs of this highly vulnerable population. As such, the primary objective of this study was to characterize the content of an international sample of Twitter messages related to pregnancy and mental health during the period of the initial wave of COVID-19, from March to June 2020.

## 2. Method

### 2.1. Ethics Approval

As Twitter can be considered a public sphere where users can and expect others, including strangers, to see and follow their actions, it is not considered necessary to obtain ethics approval for the analysis of tweets, as each tweet is considered publicly available information. It is recognized that in the setting where a user expects a certain amount of privacy (e.g., private group messages, closed forums, and invite-only support groups), ethical approval to use these data would be necessary; these were not consulted in the current work. Consequently, no identifiable information nor direct quotes of tweets permitting the identification of users was used in the publication of this manuscript; any identifiable information was redacted or not included. Throughout this article, representative tweets were created as examples of tweets from our dataset.

### 2.2. Search Strategy

The search was conducted on the Twitter platform using a predefined list of keywords under the hashtag format pertaining to three categories: COVID-19, pregnancy, and mental health (see Table 1). To be included in the search, a tweet had to contain at least one hashtag from each category, be written in English, and have been tweeted between 1 March and 31 May 2020. Categories were chosen in such a way as to maximize the number of pertinent tweets about pregnancy and mental health while limiting any irrelevant tweets.

Tweets were gathered using the Python library called GetOldTweets3 [32]. Only tweets using 8-bit transformation code encoding format (UTF-8) compatible fonts were included, effectively leaving un-encodable fonts and characters out of this dataset. Tweets were then geolocated using Twitter’s application programming interface (API), then applied through a Python script using the Tweepy library (accessible at: https://www.tweepy.org). In all, 175 accounts associated with the collected tweets were geolocated using this technique while the remaining 17 were manually coded given that the geolocation option was not enabled by the owners of those accounts. The manual geolocation was independently verified by two researchers, while the geolocations of users that were either unsure or not marked were left as unknown.

### 2.3. Data Analysis

A three-step approach was used to analyse tweets. Firstly, a sentiment analysis was executed using the Valence Aware Dictionary and sEntiment Reasoner (VADER), a lexicon-based sentiment analysis tool with a reported high accuracy, specifically with respect to social networking platform settings such as tweets [33,34]. This tool is based on the Python computer programming language and is distributed as a Python library. The compounded score of polarity, which indicated if a tweet had a generally negative (score from −1 to −0.05), neutral (score from −0.05 to 0.05) or positive tone (score from 0.05 to 1), was then generated. Once the VADER library attributes a score to each tweet, a global mean score is generated using the mean function of Microsoft Excel. The same approach was used to generate the median scores for each group by using the median function.

Secondly, accounts linked to the gathered tweets were manually sorted and peer-reviewed into four major categories identifying the type of author: (1) individuals, in which the authors used a personal Twitter account; (2) health professionals/researchers, where the authors were promoting/recruiting for research or services from a professional account; (3) non-profit organizations, including non-governmental and community organizations, and (4) companies, in which the authors were promoting/selling products or services for monetary gains. This approach allowed us to identify the needs of individuals (targeted group) and compare them to resources and support offered by the other identified entities mentioned.

Thirdly, the themes presented in the tweets were manually identified and validated through peer-review. The peer-review process consisted of the two first authors independently noting recurrent themes for all tweets and then comparing their work. Thematic analysis focused on the mental health aspects of tweets. When a discrepancy occurred between identified themes, the two authors discussed to find an agreement.

## 3. Results

In total, 192 tweets were analyzed (see Figure 1). Most tweets (91) were from the United States of America, while 47 tweets were from Great Britain and 20 from Canada. Other tweets were from Australia (6), Ireland (6), Nigeria (5), India (5), Italy (1), Qatar (1), Turkey (1), Spain (1), and Lebanon (1). The geolocation of the five remaining tweets could not be verified. The 192 tweets were then classified by author type (see Table 2). In total, 51 tweets were from individuals, 48 from health professionals/researchers, 56 from non-profit organizations, and 37 were tweets from companies. Among these, private healthcare providers (11), media such as news website and journals (9), mental health providers (8), educators (4), and others (5), such as cookbook sellers, marketing cellphone applications, and online pharmacy companies were identified.

### 3.1. Sentiment Analysis

Throughout the three months (13 weeks) of data sampling, the average sentiment score for the whole dataset had a global negative score of −0.05. Companies (*n* = 37) and individuals (*n* = 51) had a median negative sentiment analysis score of −0.29 each, while non-profit organizations (*n* = 56) and health professionals/researchers (*n* = 48) had a median neutral sentiment analysis score of 0.0 (see Figure 2).

While both companies and individuals generally expressed the same type of keywords (i.e., stress, depress*, anx*, alone, mental*, support), the context of the tweets and the themes present were wildly different. Individuals on Twitter tended to tell their personal stories, share their feelings, and often tried to help others in similar situations through normalization of negative feelings during this difficult time: “I can’t believe I will deliver my first #baby alone in the hospital because of #covid19 #Stress #Anxiety #Exhausted #MentalHealth #sad #pandemic”; “I’m #pregnant and really scared because of this #coronavirus. If you are in the same situation, tweet back and let’s support each other #pregnancy #baby #covid19 #stress #worried.”

Meanwhile, companies used the terms in a manner that would allow them to gather more traction on the Twitter platform. While they did offer help to women suffering from anxiety, depression, or other health conditions, they did so in a manner that aimed to increase exposure for the services they offered, often grouping unrelated conditions together in a post and putting a COVID-19 related hashtag. Of the 37 tweets identified as coming from a company account, 22 tweets were directly trying to sell a service or linking their company’s website to attract more people: “Don’t be scared to reach for support during your #pregnancy, especially during #COVID19, you are not #alone #CallUsNow or visit our #website #MentalHealth #Stress #Depression #Anxiety #Postpartum #GetHelp #Coronavirus #StayHome #Isolation #Quarantine.”

The other 15 tweets coming from company-related accounts typically offered resources or links to news articles on safety measures and advice on how to deal with pregnancy and mental health during the pandemic: “This #pandemic is a difficult time for everyone, but #pregnancy is a vulnerable period in a #woman’s life. If you are #pregnant and #anxious because of #covid19, we answer most common questions in the #article below.”

### 3.2. Common Themes

Among the individual tweets, the most common themes were about anxiety and stress related to COVID-19 during pregnancy. Among these tweets, three main themes emerged: stress about being isolated and alone during the pregnancy and the birth, depressive symptoms and sleep difficulties related to stress, (see Figure 3). Almost half of tweets from companies, organizations, and health professionals were about the stress and depressive symptoms that pregnant women may experience during the pandemic, as well as the services offered to help relieve these symptoms. However, few tweets were about being isolated or sleep difficulties.

#### 3.2.1. Stress

Half of the tweets from individuals (27/51; 52%) expressed feeling stress due to being pregnant during the COVID-19 pandemic. In these tweets, pregnant women conveyed the desire for this pandemic to end: “Being pregnant during a global pandemic is so stressful, I can’t wait for it to be over and enjoy the rest of my pregnancy #stress #covid19pandemic #pregnancy.” Others mentioned experiencing stress because of the global pandemic: “This is my first #pregnancy and I am so #stressed, being in third trimester during this #covid-19 isn’t helping.’” As for companies, health professionals, and non-profit organizations, almost half of the tweets (67/140; 47%) were about supporting pregnant women during the pandemic, either by proposing a service to them or sharing information and resources: “Are you #Pregnant during this #covid19? Feeling #stressed and overwhelmed is normal during this #pandemic, check out our tips to manage #stress #maternalmentalhealth #pregnancy #anxiety,” often linking a website leading to resources and contact information to the organization or service provider.

#### 3.2.2. Isolation

Social distancing and isolation are themes that were present among individuals’ tweets, where 10 out of 51 tweets (19%) mentioned the difficulty of being socially distanced from friends and family during their pregnancy: “If you know someone pregnant, tell them you are thinking about them, it’s not easy being away from everyone during this pandemic.” The apprehension of giving birth alone in the hospital due to a reduced number of potential visitors was also a concern: “The thought of giving birth alone is so stressful #pregnancy #COVID19 #anxiety,” as was not being able to go to medical appointments: “My city is now in lockdown and my doctor appointment got cancelled, I feel like crying.” Companies, health professionals, and non-profit organizations mentioned isolation only in 7 tweets out of 140 (5%) “Hit the link down below for tips and advice on what you can do to cope with #isolation due to this #coronavirus” with a link leading to the company’s website. Other tweets mentioned the possible impacts of isolation on mental health: “Professionals expect an increase in #mental health difficulties such as #anxiety and #depression in pregnant women because of the #pandemic and #social isolation,” seeking to inform the public about the possible detrimental effects of isolation and bring attention to the issue.

#### 3.2.3. Depression

Individuals’ tweets mentioned being pregnant and feeling depressed in 8 out of 51 tweets (15%). “This covid-19 is horrible, I struggle with pregnancy depression and now we just started lockdown where I am #pregnant #baby #depression #coronavirus #mentalhealth.” Companies, health professionals, and non-profit organizations mentioned depression in 27 out of 140 tweets (19%). Eight tweets were from health professionals or researchers and either stated results from studies or gave general statistics about depression in pregnant women: “Before the #pandemic, around 25% of #pregnant women had mild-severed depression and 10% had moderate-severe depression #mentalhealth.” Tweets from organizations (*n* = 9) were trying to bring awareness to the issue: “#isolation and #social distancing during the #covid19 pandemic will increase #depression, #anxiety and other #mentalhealth disorders in #moms and #pregnant women.” A further 13 tweets from companies tended to offer their paid services to help cope with symptoms that may stem from isolation: “Are you #pregnant or a new #mom? Reach out to us for support if you need help managing #stress or #depression.”

#### 3.2.4. Sleep Difficulties

Individuals mentioned having sleep difficulties during their pregnancy in 4 out of 51 tweets (7%). More precisely, sleep problems seemed to be related to the pandemic and the stress that it caused: “I’m exhausted. I cannot sleep because the thought of this #covid-19 gives me anxiety. Guess I’ve sign up for many bad nights of sleep during this #pregnancy #anxiety #stress.” As for companies, health professionals, and non-profit organizations, only 2 tweets out of 140 (1.4%) acknowledged sleep difficulties among pregnant women during the pandemic. The first one was from a private healthcare provider linking a news’ article from their website: “the pandemic is causing sleep disturbance in new parents and pregnant women,” while the second tweet was from a health professional linking to research about symptoms of COVID-19 in pregnant women.

## 4. Discussion

This study investigated 192 tweets from an international sample concerning the mental health of pregnant women during the early months of the COVID-19 pandemic, specifically from 1 March to 31 May 2020. Using the Twitter platform as insight into the reality of women currently living in this pandemic can inform us of the challenges women face and the support and resources to which they may have access.

Results from the sentiment analysis found that individuals and companies tended to use the same negative tonality while posts from organizations and health professionals/researchers were neutral. Interestingly, while both individual and company accounts used the same type of words such as anxiety, stress, and depression, the context in which they were used was extremely different. Within individual accounts, 18 tweets were from personal experiences about their situation during the COVID-19 pandemic and typically used negatively toned language as their struggles involved a form of stress, anxiety, depressive symptoms, isolation, and sleep difficulties. Meanwhile, companies used the same terminology overall, thus attributing to them a negative score similar to that of the individuals; however, the context of those tweets was either an offer of services or resources to help with the aforementioned issues experienced by women.

Results from the thematic analysis revealed that the main challenge lived by pregnant women during the pandemic was stress and coping with it. Tweets from individuals expressed other difficulties surrounding their mental health during this pandemic, such as troubles with isolation, sleep difficulties, and depressive symptoms.

### 4.1. Stress

Perinatal maternal stress can have many negative consequences on the mother and the offspring [10]. Major events such as the COVID-19 pandemic can exacerbate feelings of stress and uncertainty, especially in vulnerable populations [35]. An analysis of the individuals’ tweets revealed that an overwhelming number of mothers expressed a significant amount of stress surrounding their pregnancy related to the global pandemic. These results align themselves with those from Corbett and colleagues [18], in which the pandemic was found to exacerbate levels of stress felt by pregnant women. Tweets did not specify the sources of the experienced stress, but as identified by Corbett and colleagues [18], sources of stress were typically found to be anxiety about their own health as well as that of their children and their unborn child. The depth of this dataset of tweets is limited and as such, we cannot attribute the true source of stress experienced by these women, apart from the global pandemic. As for the non-individual accounts (companies, organizations, and health professional/researchers), results found that they seemed to fulfill the need expressed by tweeting individuals, by explicitly offering resources (under the form of infographics and articles) and services. These tweets also brought awareness about the special circumstances that pregnant women face during the global pandemic.

### 4.2. Isolation

Currently, the primary strategy to prevent the spread of COVID-19 is social distancing [16]. Studies found that social distancing measures significantly reduced social contact and increased feelings of loneliness in the general population [36,37,38]. Different levels of social distancing are in place from maintaining a set distance between individuals (i.e., 2 m of distance) to a “lockdown” involving closures of non-essential services and population-wide isolation. Being isolated for a long period of time can increase stress [20] and results for the thematic analysis revealed that isolation was one of the reasons that individuals expressed distress over time. As seen in McLeish and Redshaw’s study [21], pregnant women are more likely to feel the effects of isolation, thus making them especially vulnerable to the pandemic precautionary measures. Separation from loved ones caused important distress, as well as missing important medical appointments due to the precautionary measures of social distancing and isolation. Our results also found that the possibility of giving birth alone in the hospital was distressing to pregnant women. Many hospitals were overcrowded during the spring peak of the pandemic, had limited supplies with which to treat patients, and had staff who were overworked [39]. Consequently, the pandemic caused hospitals to refuse patients who needed care that were not related to COVID-19, such as was the case for pregnant women [39]. In a survey asking 592 American pregnant women how their birth plan changed during the pandemic, 45.2% responded having to change some aspect of it because of the changes and restrictions brought forth by COVID-19. Among these women, 367 reported a fear of giving birth alone in the hospital because of room restrictions due to the virus (i.e., fewer rooms available as well as fewer or no visitors allowed in the room during and after labour) which supports our findings [39]. As for the non-individual accounts (companies, organizations, and health professional/researchers), only 5% (7/140) of tweets acknowledged difficulties related to isolation in pregnant women. It is alarming that 19% of tweets from individuals reported having some difficulties due to the confinement and only 5% of non-individual accounts recognized this as a possible issue, demonstrating a rift between the services offered and the demand expressed by the users. Not addressing this problematic may lead to depression and a lower mood in pregnant women [21,23]. A solution could be the creation and advertisement of social support networks delivered online, which has been shown to help pregnant women with maternal well-being, lower depression/anxiety, and improve social support [24,25].

### 4.3. Depression

Findings revealed that 15% (8/51) of analyzed tweets from individuals mentioned depressive symptoms. In non-pandemic times, about 1 in 5 women will present some form of mental health problem during their pregnancy, more often anxiety and depression [40]. Rates of perinatal depression in economically developed countries, where most of the dataset from the present study came from, range from 7% to 15% [41]. In a survey with 1987 Canadian pregnant women, 37% reported depressive symptoms, which represents an elevation compared to pre-pandemic times [42]. Many of the analysed tweets mentioned that the pandemic played a role in the development or maintenance of their depressive symptoms. Non-individual accounts (companies, organizations, and health professional/researchers) mentioned depression in 27 tweets out of 140 (19%), which is similar to the 15% of tweets from individuals reporting depressive symptoms, indicating that depression during pregnancy was acknowledged by non-individual accounts who offered solutions and resources for individuals.

### 4.4. Sleep Difficulties

In the individuals’ tweets, 7% (4/51) reported sleep difficulties. More precisely, individuals reported difficulties with falling asleep, which seemed directly linked to the stress caused by the pandemic. Confinement can negatively affect sleep, probably due to low levels of activity and higher levels of stress [9,43]. Moreover, many individuals experienced a change in their routine (e.g., working from home, caring for children staying home because of closed daycares and schools, not being able to engage in rewarding activities such as visiting friends or family) and increased levels of uncertainty and worries related to the global pandemic, which can impact sleep quality [9,43]. A reduction in sleep quality is usually expected during pregnancy, but if sleep quality worsens to a point of affecting a woman’s functionality or quality of life, an intervention may be required given that poor sleep quality is predictive of depression in pregnancy [44,45]. As for companies, health professionals, and non-profit organizations, only 1.4% (2/140) of tweets mentioned sleep difficulties. Interestingly, a recent study found that 45.5% of pregnant women experience sleep difficulties [44] and given the fact that disturbed sleep in pregnant women is exacerbated by the stress of the global pandemic [9], it is surprising that so few non-individual accounts acknowledge this as a possible issue requiring solutions and resources.

### 4.5. Limitations and Future Directions

As the dataset spanned the first three months of the pandemic, it included a limited sample size of 192 tweets, which gave us a narrow view of the effects of the pandemic on pregnant women. A follow-up study should be made of tweets from June to December 2020 to analyze the progression of this issue.

A limitation of this study is the use of data uniquely gathered through Twitter, therefore missing an important portion of the population. It can be assumed that many women would use social media platforms other than Twitter. As such, the next logical step for a future study would be to investigate other social media platforms.

As this study was based on tweets, it lacks validity measures as no questionnaire or self-reported measures were used. A future study could use Twitter data and self-reported measures in conjunction for both individuals and health professionals, researchers, and non-profit organizations, to assess the needs of pregnant women and the perceived available support and resources during the COVID-19 pandemic.

## 5. Conclusions

Albeit limited, our sample can provide insight on how the pandemic has affected women during and after their pregnancy compared to pre-pandemic data on perinatal mental health problems. Tweets mentioned how the pandemic not only affected but compromised the support available to them during pregnancy. From the available tweets, we observed that non-individual accounts gave adequate support and services to women that expressed concerns around depressive symptoms and stress. However, isolation and sleep difficulties seemed to be lacking in support, resources, and acknowledgment from non-individual accounts. While Twitter can be recognized as a platform able to offer social support and information on mental health [27,28], our results highlight the need for companies, organization, and healthcare professionals/researchers to fill this void on social media, through offering support and resources on a greater number of mental health issues for pregnant women. More precisely, isolation and sleeping difficulties need to be addressed while continued support regarding stress and depressive symptoms is required, especially during this global pandemic.

## Figures and Tables

**Figure 1 ijerph-18-00393-f001:**
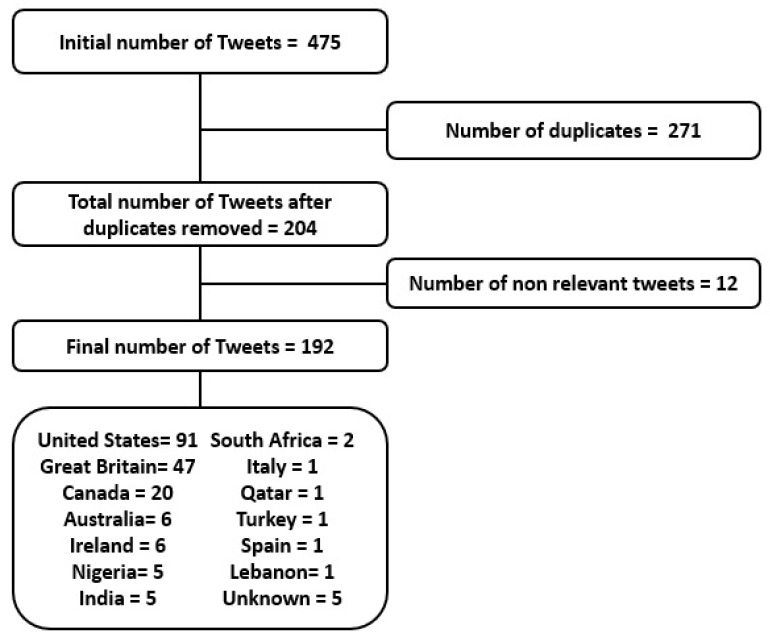
Tweets selection process flowchart.

**Figure 2 ijerph-18-00393-f002:**
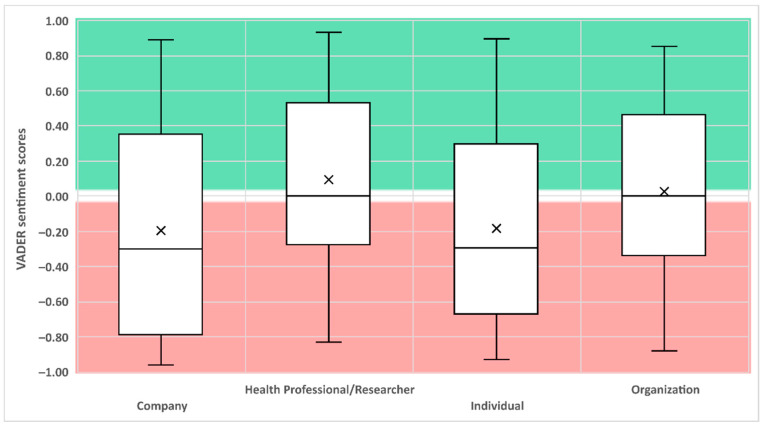
The Valence Aware Dictionary and sEntiment Reasoner (VADER) scores of the four identified categories. Note: Score from −1 to −0.05 are considered having a negative sentiment. Scores from −0.05 to 0.05 are considered neutral. Scores from 0.05 to 1 are considered positive. The middle line represents the median while the × represents the mean.

**Figure 3 ijerph-18-00393-f003:**
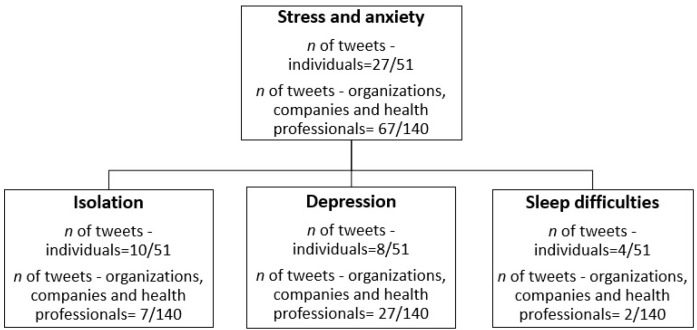
Thematic analysis and tweets repartition.

**Table 1 ijerph-18-00393-t001:** Search keywords and categories.

Hashtags Related to COVID-19	Hashtags Related to Pregnancy	Hashtags Related to Mental Health
coronavirus	pregnancy	depressed
covid-19	pregnant	sad
covid19	maternity	worried
virus	obstetrics	anxious
SARS-CoV-2	baby	stress
Pandemic	birth	anxiety
2019 novel coronavirus	postpartum	worry
covid19pregnancy	postnatal	depression
coronaviruspregnancy	prenatal	mental health
SARSCOV2	prepartum	stressed
covid_19pregnancy	peripartum	alone
pregnancycoronavirus	laboranddelivery	
pregnantcovid-19	covid19pregnancy	
pandemicpregnancy	coronaviruspregnancy	
	covid_19pregnancy	
	pregnancycoronavirus	
	pregnantcovid-19	
	postpartumhealth	
	postpartumsupport	
	pandemicpregnancy	
	perinatalmentalhealth	

**Table 2 ijerph-18-00393-t002:** Descriptive count of tweets by country and author type.

Country	Total Number of Tweets	Individual	Health Professional/Researcher	Non-Profit Organization	Company
United States	91	19	22	28	22
Great Britain	47	17	5	20	5
Canada	20	3	12	4	1
Australia	6		1		5
Ireland	6	1	3	2	
Nigeria	5		5		
India	5	2		1	2
South Africa	2	2			
Italy	1	1			
Qatar	1				1
Turkey	1			1	
Spain	1	1			
Lebanon	1	1			
Unknown	5	4			1
Total	192	51	48	56	37

Note: the countries were sorted to the number of tweets from each country from the highest to the lowest.

## Data Availability

The data presented in this study are available on request from the corresponding author. The data are not publicly available in order to retain the anonymity of the tweeters.

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
