# Peer review of "Feeling the Void: Lack of Support for Isolation and Sleep Difficulties in Pregnant Women during the COVID-19 Pandemic Revealed by Twitter Data Analysis"

_ijerph, 2021, doi:10.3390/ijerph18020393_

Round 1

Reviewer 1 Report

The idea and design is new. But the result is simple and the sample size is small. I think maybe it can be published as a report.

Author Response

The idea and design is new. But the result is simple and the sample size is small. I think maybe it can be published as a report.

We thank the reviewer for the thoughtful comment. We addressed the limited sample size as a limitation in an added section (see line 403).

Reviewer 2 Report

Dear authors,

I would like to congratulate you on the relevance of the present study and the interesting experimental design, using a social media platform to inform about the expression of sentiments during such challenging times. In my opinion the overall appreciation of the manuscript is fairly good, considering the proper rationale and framework for the research question. The provided elements throughout the manuscript are relatively clear and informative for the potential reader. However, I have some concerns or comments for your consideration, as following:

Abstract

Line 19: Could authors provide a clearer information about how the thematic analysis procedures were performed?

Line 27: the following keywords “Pregnant Women”, “COVID-19” and “Twitter” are repeated from the title. Please consider changing them, in order or expand the search possibilities in the future.

Introduction

I find it well written, with a fluent narrative, providing the potential reader with a proper framework for understanding the rationale supporting the research question and study design.

Methods

The use of a netnographic approach for the experimental design is very interesting and well-fitted for the study objectives, even more considering the social distancing and isolating context. In my opinion, this section must be very clear and informative, particularly regarding the reliability and validity of all procedures. For example, it is not clear to me how the peer-review process, as presented, is enough to support findings; nor the rationale for the categories identifying the type of author.

Line 143: What was the rationale for the categories identifying the type of author?

Line 151: Please explain this peer-review process. Who were the reviewers? Was any control quality of data performed (e.g., intra and inter observer reliability)?

Results

Outcomes are appropriate for the study design and the graphic elements included in the manuscript allow the potential reader to have a clear and complementary interpretation of the observed results.

Lines 246–248: Please change to italic, in accordance with previous quotes.

Lines 251–252: Please change to italic, in accordance with previous quotes.

Lines 253–255: Please change to italic, in accordance with previous quotes.

Lines 257–258: Please change to italic, in accordance with previous quotes.

Discussion The authors are cautious in analyzing and discussing the observed results. However, every study has its own limitations and authors should clearly recognize the present study weaknesses and suggest further studies or practical implications. The concept of mental health is of major importance, particularly in such an unprecedent event as this Pandemic and its possible effects in a specific set of women. I suggest that this concept can be presented or clarified in the manuscript. Self-reported measures can be very useful (Antunes et al., 2020), but alternative tools for assessing it should be here discussed, emphasizing the relevance of social media in the characterization of such symptoms.

Antunes, R.; Frontini, R.; Amaro, N.; Salvador, R.; Matos, R.; Morouço, P.; Rebelo-Gonçalves, R. Exploring Lifestyle Habits, Physical Activity, Anxiety and Basic Psychological Needs in a Sample of Portuguese Adults during COVID-19. Int. J. Environ. Res. Public Health 2020, 17, 4360.

Line 338: This finding seems to be worthy of a further discussion.

Line 361: This was demonstrated in the Antunes et al. (2020) study, where female participants revealed changed sleeping habits during the COVID-19 first wave (unsatisfied with sleep quality and unusual sleep duration), when compared to men.

Line 371: I strongly advice for the need of a higher emphasis on study limitations and practical implications, as presented in the abstract.

Author Response

Dear authors,

I would like to congratulate you on the relevance of the present study and the interesting experimental design, using a social media platform to inform about the expression of sentiments during such challenging times. In my opinion the overall appreciation of the manuscript is fairly good, considering the proper rationale and framework for the research question. The provided elements throughout the manuscript are relatively clear and informative for the potential reader. However, I have some concerns or comments for your consideration, as following:

We thank the reviewer for their feedback and manuscript appreciation.

Abstract:

Line 19: Could authors provide a clearer information about how the thematic analysis procedures were performed?

Thank you for this comment, with the limit of 200 words for the abstract, we feel that this information would be more pertinent and useful for the reader in the methods sections as per your comment (see below on line 157).

Line 27: the following keywords “Pregnant Women”, “COVID-19” and “Twitter” are repeated from the title. Please consider changing them, in order or expand the search possibilities in the future.

Thank you for this observation, the keywords have been changed accordingly. Line (27) “Pregnancy; Mental Health; SARS-CoV-2; Women’s Health; Social Media

Introduction:

I find it well written, with a fluent narrative, providing the potential reader with a proper framework for understanding the rationale supporting the research question and study design.

Thank you for taking the time to write this feedback, it is much appreciated.

Methods:

The use of a netnographic approach for the experimental design is very interesting and well-fitted for the study objectives, even more considering the social distancing and isolating context. In my opinion, this section must be very clear and informative, particularly regarding the reliability and validity of all procedures. For example, it is not clear to me how the peer-review process, as presented, is enough to support findings; nor the rationale for the categories identifying the type of author.

Line 143: What was the rationale for the categories identifying the type of author?

Thank you for this question. When we conducted the data analysis, we quickly noted that different types of authors seemed to have differing intent when tweeting. Taking this into consideration, we could not compare all tweets (from individual, health professionals, companies, non-profit organization) together without first dividing them into categories, given that the goals/targeted population/purpose of the tweets were not homogenous. A sentence was added to clarify why this approach was chosen (see line 157).

Line 151: Please explain this peer-review process. Who were the reviewers? Was any control quality of data performed (e.g., intra and inter observer reliability)?

Thank you for your comment. We expanded on the process of peer-review for the identification of themes (See line 161-165): “The peer-review process consisted of the two first authors independently noting recurrent themes for all tweets and then comparing their work. Thematic analysis focused on the mental health aspects of tweets. When a discrepancy occurred between identified themes, the two authors discussed to find an agreement.”

Results:

Outcomes are appropriate for the study design and the graphic elements included in the manuscript allow the potential reader to have a clear and complementary interpretation of the observed results.

Lines 246–248: Please change to italic, in accordance with previous quotes.

Lines 251–252: Please change to italic, in accordance with previous quotes.

Lines 253–255: Please change to italic, in accordance with previous quotes.

Lines 257–258: Please change to italic, in accordance with previous quotes.

Thank you. These changed have been made in the manuscript.

Discussion:

The authors are cautious in analyzing and discussing the observed results. However, every study has its own limitations and authors should clearly recognize the present study weaknesses and suggest further studies or practical implications. The concept of mental health is of major importance, particularly in such an unprecedent event as this Pandemic and its possible effects in a specific set of women. I suggest that this concept can be presented or clarified in the manuscript. Self-reported measures can be very useful (Antunes et al., 2020), but alternative tools for assessing it should be here discussed, emphasizing the relevance of social media in the characterization of such symptoms.

Thank you for these thoughtful comments. We added a section in the discussion with limitations and future studies (see line 403); we feel that this does indeed strengthen our manuscript.

Line 338: This finding seems to be worthy of a further discussion.

We are grateful for this suggestion. We expanded on this finding in the discussion (see line 352) and the possible practical implications.

Line 361: This was demonstrated in the Antunes et al. (2020) study, where female participants revealed changed sleeping habits during the COVID-19 first wave (unsatisfied with sleep quality and unusual sleep duration), when compared to men.

Thank you for the thorough comment and article suggestion. We found the article extremely interesting and have added the reference in the text to further support this point (see line 375 and 379).

Line 371: I strongly advice for the need of a higher emphasis on study limitations and practical implications, as presented in the abstract.

Thank you, we have added a section in the discussion about our limitations and future directions (see line 403).

Reviewer 3 Report

Authors discuss the impact of Covid-19 on pregnant women. 

The topic is very interesting and the approach is original. Indeed Twitter data have been used to derive the woman's feelings.

The article is well written and the analysis has been correctly conducted. 

Related work section should be better discussed by including other works where twitter data are used to derive topics or hidden knowledge (e.g. https://doi.org/10.1108/IJWIS-11-2017-0081 , https://doi.org/10.1080/21645515.2020.1714311 ). Moreover, I suggest to include automatic topic detection in future works.

Authors should describe how the average sentiment score has been obtained. Is it automatically evaluated by one of the libraries they have used? If so, they should detail how this score is evaluated.

Moreover it is not clear why four categories of users have been identified (Individual, Health professional, etc.) and how these classes are relevant for the analysis. Authors should better highlight this point.

Author Response

Authors discuss the impact of Covid-19 on pregnant women. 

The topic is very interesting and the approach is original. Indeed Twitter data have been used to derive the woman's feelings.

The article is well written and the analysis has been correctly conducted. 

Related work section should be better discussed by including other works where twitter data are used to derive topics or hidden knowledge (e.g. https://doi.org/10.1108/IJWIS-11-2017-0081 , https://doi.org/10.1080/21645515.2020.1714311 ). Moreover, I suggest to include automatic topic detection in future works.

Thank you for this suggestion, we will consider the automatic topic detection in the future. Thank you for these great references, they will be a great addition to the introduction section of this manuscript (see line 93-98).

Authors should describe how the average sentiment score has been obtained. Is it automatically evaluated by one of the libraries they have used? If so, they should detail how this score is evaluated.

Thank you for this question. Briefly, the VADER library was used to generate a sentiment score for each tweet using a lexicon-based approach. Then, the median score was generated using the median function from Excel for the global score of the data set. The same approach was used for each of the groups. The VADER library does not automatically produce medians or means.

We added more information on this in the Data Analysis section (see line 147-150).

Moreover it is not clear why four categories of users have been identified (Individual, Health professional, etc.) and how these classes are relevant for the analysis. Authors should better highlight this point.

Thank you for this comment, another reviewer had a similar stream of thought. When we did the data analysis, we quickly noted that different type of authors had different goals when tweeting. Taking this into consideration, we could not compare all tweets (from individual, health professionals, companies, non-profit organization) together without first dividing them into categories, since the goals/targeted population/purpose of the tweets were not homogeneous. A sentence was added to clarify why this approach was chosen (see line 157).